# Clinical Evaluation of the Efficacy of Femtosecond Laser-Assisted Anterior Lamellar Keratoplasty

**DOI:** 10.3390/jcm12031158

**Published:** 2023-02-01

**Authors:** Adam Wylęgała, Anna M. Roszkowska, Joanna Kokot, Dariusz Dobrowolski, Edward Wylęgała

**Affiliations:** 1Health Promotion and Obesity Management Unit, Department of Pathophysiology, Faculty of Medical Sciences, Medical University of Silesia, 40-055 Katowice, Poland; 2Ophthalmology Department, Railway Hospital, 40-760 Katowice, Poland; 3Ophthalmology Clinic, Department of Biomorphological Sciences, University of Messina, 98124 Messina, Italy; 4Ophthalmology Section, Faculty of Medicine and Health Sciences, Andrzej Frycz Modrzewski Krakow University, 30-705 Kraków, Poland; 5Chair and Clinical Department of Ophthalmology, II School of Medicine with the Division of Dentistry in Zabrze, Medical University of Silesia, 40-760 Katowice, Poland; 6Ophthalmology Clinic of St. Barbara Hospital, 42-200 Sosnowiec, Poland

**Keywords:** anterior lamellar keratoplasty, femtosecond laser, optical coherent tomography, keratoconus, corneal dystrophies, corneal degenerations

## Abstract

Background: To evaluate the clinical outcome, efficacy, and safety of femtosecond laser-assisted anterior lamellar keratoplasty. Material and Methods: In this prospective study, 21 males and 10 females aged from 15 to 62 years (mean 38.5) with different pathologies of the anterior corneal layers were enrolled for anterior lamellar keratoplasty using femtosecond laser VisuMax (Carl Zeiss, Germany). All patients were examined for uncorrected (UCVA) and best-corrected (BCVA) distance and near visual acuity, astigmatism, endothelial cell density, corneal thickness, and intraocular pressure. These examinations were performed before transplantation, at hospital discharge, and after 3 and 12 months. The mean follow-up time was 65.36 ± 28.54 months. Results: A statistically significant improvement of both UCVA and BCVA for distance and near vision was registered. BCVA improved from 0.11 preoperatively to 0.168 (*p* = 0.03), 0.267 (*p* < 0.01), and 0.472 (*p* < 0.01) on the hospital discharge day, three months, and 12 months respectively. There were no statistically significant differences in astigmatism, intraocular pressure, endothelial cell density, and corneal pachymetry as compared to preoperative and postoperative values. Six patients (19%) had a graft failure with a rate of 33% at 26 months. Conclusions: Femtosecond laser-assisted anterior lamellar keratoplasty is a safe and effective surgical method, providing satisfactory graft survival rates.

## 1. Introduction

In corneal transplantation, the most novel surgical techniques aim to transplant only the affected part of the cornea to minimize the risk of graft rejection and efficiently use the donor tissue, which may be used in more procedures [1]. Anterior lamellar keratoplasty replaces the anterior part of the corneal stroma, leaving a healthy Descemet membrane and endothelium layer. Indications for this procedure include corneal pathologies such as anterior dystrophies and degenerations [2], corneal scars [3], keratoconus [4], and corneal ulcers at risk of perforation [5]. The most critical and challenging part of lamellar surgery is separating corneal tissue at the appropriate depth (which depends on the extent of corneal pathology) to leave a plain bed for the corneal graft. The most popular method is the big-bubble technique of Anwar and Teichman [6], which allows the smooth Descemet membrane to be exposed using air. However, the use of this procedure is limited by the long learning curve and the risk of Descemet membrane perforation, which can require a switch to penetrating keratoplasty [7]. Therefore, automated cutting processes using microkeratomes and lasers are suitable to achieve high precision and smoothness of the corneal interfaces during the lamellar keratoplasty. In 2005, for the first time, a femtosecond laser was used experimentally for anterior lamellar keratoplasty [8]. Such a laser generates short pulses of energy that change tissue to plasma in a process called photodisruption. This reaction is connected with the generation of gases that pull the tissue apart. The strengths of femtosecond laser for corneal surgeries include the ability to create precise and accurate incisions, which can improve the outcomes of the surgery. The laser can also reduce surgical time and complications [9]. The most crucial advantage of using femtosecond lasers to perform corneal cuts is that the depth and pattern of the cut can be selected, which allows the individualization of each procedure according to the cornea changes and the surgeon decision. The aim of this study was to evaluate the efficacy and safety of femtosecond laser-assisted anterior lamellar keratoplasty (FALK) in patients with different corneal pathologies.

## 2. Materials and Methods

This prospective study was approved by The Bioethical Committee at the Silesian Medicine University (resolution number KNW/0022/KB1/103/I12), and all experiments were performed in accordance with relevant guidelines and regulations. Patients with pathologies of the anterior layers of the cornea suitable for FALK were selected. The inclusion criteria considered pathological changes in the anterior part of the cornea, best-corrected visual acuity ≤0.2, minimal corneal thickness >350 µm, and the willingness of patients to participate in a study. Patients with changes in endothelial cells and Descemet’s membrane, post infections, diagnosed glaucoma, non-complainant, and pregnant were excluded from the study. Informed written consent was obtained from all patients who agreed to participate in the study. In the case of minors, permission was given by legal guardians. The authors excluded patients with severe corneal opacities by performing thorough preoperative evaluations, including clinical examination and imaging studies. The surgical procedures were performed between December 2010 and January 2015 at the Ophthalmology Clinic of St. Barbara Hospital, Sosnowiec, Poland. Patients were examined before the surgery, on the day of hospital discharge (around the seventh-day post-keratoplasty), and after 3 and 12 months. Successively, all subjects were followed for graft failure, which was defined as rejection, inflammation requiring regrafting, recurrence of diseases, or loss of graft clarity.

The main outcome measures were (UCVA) and (BCVA) for far and near, astigmatism, endothelial cell density, corneal thickness, and intraocular pressure.

### Surgical Technique

A VisuMax 500 Hz femtosecond laser (Carl Zeiss, Jena, Germany) was used for lamellar cutting of both the donor’s and recipient’s corneas. The parameters for the cutting process were entered into the laser system’s memory before the procedure (Table 1).

The placement of corneal changes and preoperative pachymetry determined the depth of lamellar cut. The lamellar cut was made parallel to the anterior surface (Figure 1).

Topical antibiotic drops (Vigamox, Alcon, Forth Worth, TX, USA) were administered, Tears Naturale (Alcon), and topical 0.1% Dexamethasone (Polfarma, Warszawa, Poland) slowly tapered over 3–4 months. Sutures were removed during the follow-up controls in patients with high astigmatism exceeding 4.0 diopters starting from 6 months post-op, and hybrid contact lenses were tried (Figure 2). All patients had the sutures removed at 12 months post-op. 

All patients underwent a slit-lamp examination with photographic documentation and were examined for uncorrected (UCVA) and best-corrected (BCVA) visual acuity for near and distance vision (Snellen chart with a logMAR), intraocular pressure using Goldmann tonometry (CSO, Firenze, Italy), astigmatism with AtlasTM (Carl Zeiss Meditec Inc., Jena, Germany), and endothelial cell density using Topcon SP-3000P specular microscope (Topcon Corporation, Tokyo, Japan). Intraoperative and postoperative complications were registered. In addition, an examination using time-domain anterior chamber optical coherent tomography (AC-OCT) Visante (Carl Zeiss Meditec Inc., Jena, Germany) was performed to assess the preoperative and postoperative pachymetry, as well as the depth of the pathological stromal changes. An average pachymetry value was obtained from five measurement points: in the center and 2 mm from the center at 0°, 90°, 180°, and 270°. Statistical analysis was performed with SPSS 21(IBM, Armonk, NY, USA). The Student’s *t*-test and the Wilcoxon signed-rank test were used for statistical analysis. The Kaplan–Maier curve was used to calculate the rate of graft failure. 

## 3. Results

Thirty-two patients (66% male) aged from 15 to 62 years (mean 38.5) who had undergone anterior lamellar keratoplasty were included in the study. The main indication for the surgery was keratoconus (22 patients, 69%). Other indications included corneal scars (six patients, 19%), lattice dystrophy (one patient, 3%), Reis-Bücklers dystrophy (one patient, 3%), stromal clouding in Salzmann degeneration (one patient, 3%), and pellucid marginal degeneration (one patient, 3%). The mean values of the considered parameters evaluated at baseline, on the hospital discharge, and at three and 12 months after surgery are represented in Table 2.

Visual acuity and astigmatism

A statistically significant improvement of distance UCVA at three (range 0.01–0.4) and 12 months (range 0.01–0.8) (*p* < 0.05) was observed and the near vision improved after 12 months (range 0.5–3) after keratoplasty. The UCVA was better than the baseline in 12 patients (60%). The BCVA showed a significant improvement at each postoperative control point for far vision (*p* < 0.05) and at the three and 12-month points for near vision (range 0.5–3). After 12 months, the BCVA for distant vision increased in 28 (88%) patients and 18 (56%) patients, and the mean was superior to 0.5 (range 0.1–1). No statistically significant changes in astigmatism were observed after the procedure, although the mean values at three and 12 months post-keratoplasty were lower than the preoperative values (Table 2). 

Intraocular pressure (IOP)

No statistically significant change in the mean IOP was noted during the follow-up. Three patients had topical antiglaucoma medications added to their standard treatment on the hospital discharge due to an elevated IOP.

In addition, three patients required antiglaucoma topical drops three months after keratoplasty, and one of these patients was still under treatment at 12 months post-keratoplasty.

Endothelial cell density (ECD)

Due to the corneal swelling, it was possible to measure ECD in 11 eyes at three months of control. A statistically significant decrease of 12.9% in endothelial cell density (2433.9 to 2119.4 cells^−2^) was observed three months post-keratoplasty. Twelve months after surgery, endothelial cell density was only 0.88% (2433.9 vs. 2412.6 cells^−2^) lower than the preoperative value and was not statistically significant (Table 2). 

Pachymetry

A statistically significant increase in the corneal thickness measured by optical AS-OCT pachymetry was observed on hospital discharge, probably due to the corneal swelling. However, no significant post-keratoplasty changes were found at three- and 12-month timepoints.

Intraoperative complications

All consecutive procedures of anterior lamellar femtosecond laser-assisted keratoplasty were evaluated to assess intraoperative and postoperative complications, as shown in Table 3.

Only one corneal bed was prepared with manual trephination due to the great thinning of the cornea in an eye with pellucid marginal degeneration. No relationships were found between the occurrence of intraoperative complications and postoperative visual acuity (*p* > 0.05).

### Graft Failure Analysis

Of the 32 patients included in the study, six patients had graft failure (19%). The failure rate was 17% at ten months, 33% at 26 months, and 66% at 36 months. (Figure 3). 

Two patients had graft rejection, one patient with KC had a corneal herpetic inflammation three years after surgery, two patients required penetrating keratoplasty, and the patients with lattice and Reiss-Bucklers dystrophy had a recurrence of corneal opacity. 

## 4. Discussion

In the present long-term study, we have examined the data of patients who underwent FALK. We found that 60% of patients experienced better UCVA than preoperative, while better BCVA was noted on every postoperative visit. In addition, the failure frequency was 19%, while the failure rate was 17% at approximately one year and 66% at 36 months. 

Anterior lamellar keratoplasty is an effective surgical procedure, which has advantages over penetrating keratoplasty [10,11]. Aside from anatomical state, visual acuity (VA) is one of the most important parameters for assessing the technique’s success. The outcomes of astigmatism can be affected by the diameter of the corneal flap, as a larger flap may result in more accurate and predictable outcomes as well as selection of the corneal flap size, the surgeon’s experience and skill, and the patient’s individual characteristics [12]. Furthermore, the use of Laser in DALK for keratoconus may reduce the rate of intraoperative Descemet perforation [11]. However, problems with astigmatism and ametropia are still present in lamellar and penetrating procedures due to the use of sutures [13].

In our study, postoperative changes in astigmatism were not statistically significant. However, mean values for astigmatism on the day of leaving the hospital and three and 12 months post-surgery were higher than those reported previously [14,15]. This may be due to the shorter follow-up and the fact that self-closing cuts were not used. Our study’s varied indications for keratoplasty could also have affected the astigmatism outcomes. Keratoplasty aims to regain the optical and biophysical qualities of the normal cornea [10]. It is essential to understand that patients with ectatic diseases should have a cross linking or crescentic wedge resection procedure as the first-line treatment [16]. In our patient group, a clear host bed was achieved in the center of the cornea following cutting with the femtosecond laser. The mean postoperative pachymetry implied standard corneal thickness. However, due to great variation in the mean corneal thickness both preoperatively (range 431.4–723 µm) and postoperatively (range 377.6–753 µm), such conclusions may be misinterpretations. A proven advantage of anterior lamellar keratoplasty compared to penetrating procedures is a reduced endothelial cell density decrease, which is known to be around 11–12% during the first six months and then stabilizes [1,10,17,18]. The use of a femtosecond laser minimizes the manipulation required just above the Descemet membrane, which should reduce the endothelial cell density loss [19]. Our study noted a decrease in endothelial cell density of 12.9% at three months post-operation. This loss can be explained by problems with endothelial cell counting caused by corneal swelling after the procedure. Corneal swelling is a common complication after femtosecond corneal procedures [20]. Another explanation is that endothelial cell loss occurs due to the femtosecond laser disrupting the deep stromal area. Endothelial cell loss depends on the different femtosecond platforms used and the settings of the laser [10]. The statistically insignificant decrease of only 0.88% after 12 months confirms this assumption. Anterior lamellar keratoplasty assisted with a femtosecond laser is entirely an extraocular procedure [17,21]. The most typical complications are micro-and macro-perforations and occur infrequently [15]. In our study, the only intraoperative difficulty came from incomplete trephination (31%), which necessitated manual cutting. No statistically significant relationship was found between postoperative VA and the occurrence of intraoperative complications. Several patients suffered postoperative complications. Coster et al. indicated that the main reason for regrafting is stroma scarring and donor–host interface haze [22]. Our findings are similar to this study. 

BCVA was also similar to our results at 12 months after the procedure in a study by Abu Shousha et al. [23]. While the authors did not report any cases of graft failure, we observed failure in almost one in five participants. However, some of our patients were observed up to eight years. In the study by Chamberlain et al., BCVA at 12-months was 0.341 compared to 0.472 in our study. The authors also reported better a decrease of BCVA at the 12-month follow up compared to earlier visit [10]. The largest study on FALK (355 eyes) reported mean BCVA of 0.30 at one-year follow up. However, the initial BCVA was much higher compared to our study 0.51 vs. 1.1 [24]. 

No statistically significant increase in IOP was observed. However, three patients (9%) required additional antiglaucoma treatment. This is less than half that in other papers where transient IOP increase was observed in 18% of patients [25]. Multiple studies compared the results of FALK with other techniques. The summary is displayed in Table 4. 

There are several limitations to the current study, including a small sample size, or the fact that some of our patients were submitted to manual dissections with possible stromal scarring and worse visual prognosis. 

Table 4 compares the results of femtosecond laser assisted anterior lamellar keratoplasty (FALK) against various other techniques for the treatment of keratoconus. The table includes information on the number of eyes in each group as well as the specific results of each study.

The studies in the table show that FALK is generally comparable to other techniques in terms of visual acuity and corneal thickness but may have worse refraction outcomes. It also shows that FALK may have lower rejection rates or better visual acuity and refraction outcomes when compared to manual DALK.

It is worth noting that the number of eyes in the studies presented in Table 4 varies greatly, with the smallest study having 20 eyes and the largest study having 251 eyes. This highlights the importance of considering the sample size when evaluating the results of the studies and interpreting the findings. It is also worth noting that some of the studies have a small number of eyes, which might affect the generalizability of the results.

## 5. Conclusions

The author can conclude that femtosecond laser assisted anterior lamellar keratoplasty is a safe and effective option for the treatment of corneal disorders and may be superior to the classic DALK in certain cases. However, further studies with larger sample sizes and longer follow-up periods are needed to confirm these findings.

## Figures and Tables

**Figure 1 jcm-12-01158-f001:**
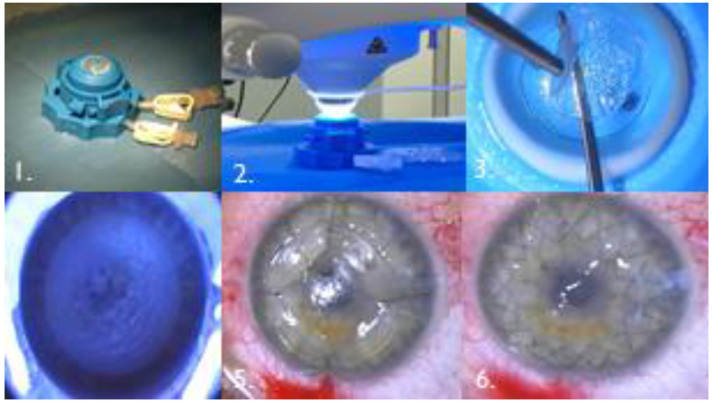
**1**. The procedure commenced with the placement of the donor tissue into an artificial anterior chamber, **2**. that was then moved under the laser’s head. **3**. After the laser had created a corneal lenticule, the flaps were separated from the rest of the corneoscleral donor tissue using a blunt spatula., if necessary, corneal scissors. **4**. The reci-pient corneal bed was then prepared in the same way. **5**. First sutures were laid in cardinal positions. **6**. The eye after continuous suture.

**Figure 2 jcm-12-01158-f002:**
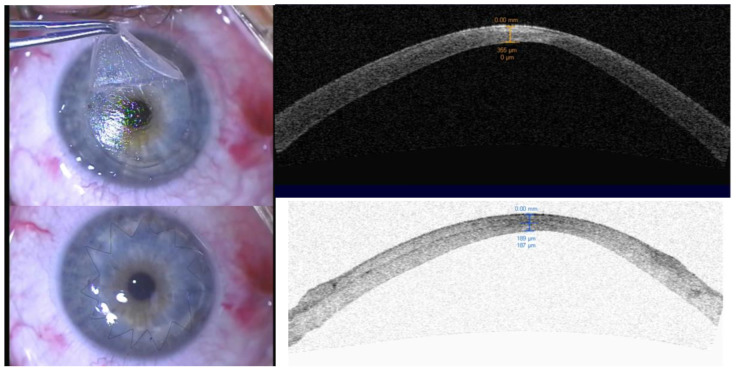
Clinical photography and the anterior segment OCT of the 44 year old female, with initial BCVA of 0.2 and 12 months post op of 0.8. The **top right** panel of the figure shows the clinical photography of the patient’s eyes during the surgery with the host flap create by the laser. The **bottom left** image shows the postoperative appearance of the eye with the single continuous suture. The **top right** panel of the figure shows the corresponding anterior segment OCT images of the patient’s eyes, which reveals the thinning of the central cornea and the irregularity of the corneal surface. The **bottom right** image shows the postoperative anterior segment OCT, which reveals the presence of a donor/host interface and a larger corneal thickness.

**Figure 3 jcm-12-01158-f003:**
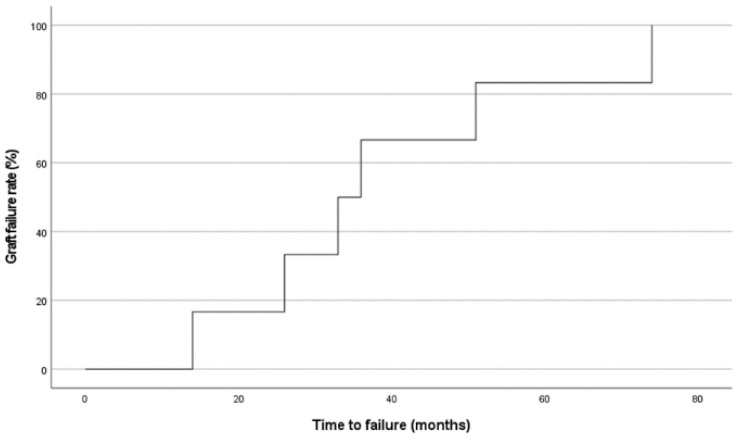
Kaplan-Meier survival curve depicting time to graft failure.

**Table 1 jcm-12-01158-t001:** The parameters for the cutting process.

Thickness of donor’s corneal flap	220–370 µm
Thickness of recipient’s corneal flap	200–350 µm
Diameter of corneal flap	7.3–8.2 mm
Side-cut angle	850–1000
Diameter of stromal bed	7.0–7.8 mm
Overlap	0.2–0.6 mm

**Table 2 jcm-12-01158-t002:** Mean values of assessed parameters *p* values are for comparison of postoperative parameter with preoperative value- Student’s *t*-test and the Wilcoxon signed-rank test.

Parameter	Preoperatively	On Day of Leaving Hospital	3 Months Post Operation	12 Months Post Operation
UCVA for distant vision decimal	0.08	0.111	0.146	0.244
		*p* = 0.08	***p* = 0.02**	***p* < 0.01**
UCVA for distant vision LogMAR	1.1	0.95	0.84	0.61
BCVA for distant vision	0.11	0.168	0.267	0.472
		***p* = 0.03**	***p* < 0.01**	***p* < 0.01**
BCVA for distant vision LogMAR	0.96	0.77	0.57	0.32
UCVA for near vision	2.688	2.333	2.146	1.679
		*p* = 0.1	*p* = 0.09	***p* = 0.02**
BCVA for near vision	2.417	2.25	1.646	1.411
		*p* = 0.38	***p* = 0.03**	***p* = 0.01**
Astigmatism (D)	4.856	5.707	4.537	4.142
		*p* = 0.23	*p* = 0.49	*p* = 0.55
IOP (mm Hg)	14.3	16.3	18.1	16.2
		*p* = 0.31	*p* = 0.14	*p* = 0.15
Endothelial cell density	2433.9	-	2119.4	2412.6
(cells mm^−2^)			***p* = 0.03**	*p* = 0.29
Pachymetry measurement (µm)	533	590.8	549.9	540.8
		***p* = 0.04**	*p* = 0.27	*p* = 0.44

BCVA—best corrected visual acuity, UCVA—uncorrected visual acuity, IOP—intraocular pressure. Statistically significant *p* values are marked in bold.

**Table 3 jcm-12-01158-t003:** Intraoperative complications.

No. of Patients	Intraoperative Complication	Results
10 (31%)	Incomplete trephination	Manual cutting
1 (3%)	Manual preparation of stromal bed	Manual trephination of recipient cornea
1 (3%)	Disturbed sequence of trephination	Smaller flap

**Table 4 jcm-12-01158-t004:** Comparison of results of studies.

Comparison of FALK against	Results in FALK Group	Number of Eyes
Big-bubble technique in keratoconus [26]	Worst BCVA and contrast sensitivity	26
Penetrating keratoplasty in keratoconus [27]	Better BCVA in patients	24
Predescemetic lamellar keratoplasty for keratoconus [28]	BCVA, CCT, and ECD insignificant, worst mean refraction	20
Manual DALK [29]	lower rejection rates, no difference in endothelial cell count between groups	251
Manual DALK [30]	better UCVA, BSCVA, and refractive and astigmatism	194
Penetrating Keratoplasty [10]	significant improvement in astigmatism, no changes in BCVA between groups	100 eyes

## Data Availability

This study was published on a preprint server. DOI: https://doi.org/10.21203/rs.3.rs-1470560/v1.

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
