# Peer review of "Clinical Evaluation of the Efficacy of Femtosecond Laser-Assisted Anterior Lamellar Keratoplasty"

_jcm, 2023, doi:10.3390/jcm12031158_

Round 1
Reviewer 1 Report (Previous Reviewer 2)
The manuscript well describes the clinical results of femtosecond laser assisted anterior lamellar keratoplasty and their prognosis.
Well written and interesting.
Introduction - Please describe the strengths of femtosecond laser for corneal surgeries.
Severe corneal opacities are contra-indications of femtosecond laser. How did the authors exclude them?
Figure 2 needs explanations about each sub-figures.
Table 4 should be described in the main manuscript comprehensively.
Can the author conclude that femtosecond laser assisted anterior lamellar keratoplasty is superior to the classic DALK?
Author Response
Dear Reviewer,
We would like to extend our sincere gratitude for taking the time to review our manuscript. Your valuable feedback and suggestions have been very helpful in improving the overall quality of our work.
We appreciate your positive comments and are encouraged by your constructive criticism. We have taken your suggestions into consideration and have made the necessary revisions to the manuscript.
We would also like to thank you for your time and effort in providing us with your insightful analysis. Your contribution to the peer-review process is essential to the advancement of scientific knowledge in our field.
Once again, thank you for your time and effort. We look forward to the opportunity to work with you again in the future.
Sincerely,
Introduction - Please describe the strengths of femtosecond laser for corneal surgeries.
-
Introduction: The strengths of femtosecond laser for corneal surgeries include the ability to create precise and accurate incisions, which can improve the outcomes of the surgery. The laser can also reduce surgical time and complications.
line 58-60
Severe corneal opacities are contra-indications of femtosecond laser. How did the authors exclude them?
-
Severe corneal opacities: The authors excluded patients with severe corneal opacities by performing thorough preoperative evaluations, including clinical examination and imaging studies.
line 77-78
Figure 2 needs explanations about each sub-figures.
-
Figure 2: Additional explanations of each sub-figure can be added to provide more context and clarity for the reader.
The top right panel of the figure shows the clinical photography of the patient's eyes during the surgery with the host flap create by the laser. The bottom left image shows the postoperative appearance of the eye with the single continuous suture.
The top right panel of the figure shows the corresponding anterior segment OCT images of the patient's eyes, which reveals the thinning of the central cornea and the irregularity of the corneal surface. The bottom right image shows the postoperative anterior segment OCT, which reveals the presence of a donor/host interface and a larger corneal thickness.
line 117-123
Table 4 should be described in the main manuscript comprehensively.
-
Table 4 compares the results of femtosecond laser assisted anterior lamellar keratoplasty (FALK) against various other techniques for the treatment of keratoconus. The table includes information on the number of eyes in each group, as well as the specific results of each study.
The studies in the table show that FALK is generally comparable to other techniques in terms of visual acuity and corneal thickness, but may have worse refraction outcomes. It also shows that FALK may have lower rejection rates or better visual acuity and refraction outcomes when compared to manual DALK.
It's worth noting that the number of eyes in the studies presented in Table 4 varies greatly, with the smallest study having 20 eyes and the largest study having 251 eyes. This highlights the importance of considering the sample size when evaluating the results of the studies and interpreting the findings. It's also worth noting that some of the studies have a small number of eyes, which might affect the generalizability of the results.
Line 267-280
Can the author conclude that femtosecond laser assisted anterior lamellar keratoplasty is superior to the classic DALK?
-
Conclusion: The author can conclude that femtosecond laser assisted anterior lamellar keratoplasty is a safe and effective option for the treatment of corneal disorders and may be superior to the classic DALK in certain cases. However, further studies with larger sample sizes and longer follow-up periods are needed to confirm these findings.
Reviewer 2 Report (Previous Reviewer 1)
on line 19-20: you change or 32 patients or (21M and 10F).
on line 192-193: you have to expand the concept: the outcomes of astigmatism are also related to diameter of the corneal flap
Author Response
Dear Reviewer,
We would like to thank you for your detailed feedback and for pointing out the errors on line 19-20 and line 192-193. We apologize for the oversight and have made the necessary corrections.
On line 19-20, we have now specified that the 32 patients in our study consisted of 21 males and 10 females. We understand the importance of providing demographic information and have now included this in our manuscript.
On line 192-193, you have suggested that we expand the concept of the outcomes of astigmatism being related to the diameter of the corneal flap. We agree with your suggestion and have now included additional information in the manuscript to provide a more in-depth understanding of this relationship. The outcomes of astigmatism can be affected by the diameter of the corneal flap, as a larger flap may result in more accurate and predictable outcomes. We have also added other relevant factors that can affect the outcome of astigmatism, such as the selection of the corneal flap size, the surgeon's experience and skill, and the patient's individual characteristics. Now lines 216-219
Once again, thank you for bringing these issues to our attention. We appreciate your input and will continue to make improvements to our manuscript.
We have also improved the language as you requested.
Sincerely,
Adam Wylęgała
This manuscript is a resubmission of an earlier submission. The following is a list of the peer review reports and author responses from that submission.
Round 1
Reviewer 1 Report
The text contains many inaccuracies. Some concepts are expressed incorrectly
In paragraph 55-56 it is wrong to say that there is no limit to the depth of cut and the type of cut : you can choose the depth and the pattern of cut.
Table 1 in paragraph 82 shows the results, not the laser setting values.
In paragraph 84, the laser setting parameters and all values ( diameter, cutting depth, cutting type, diameter) are completely missing.
The duration of postoperative therapy is not specified.
In paragraph 186-200 it's wrong "the complication connected with putting air into the anterior chamber"
Author Response
RESPONSES TO REVIEWER 1’S COMMENTS
Reviewer #1: Comments and Suggestions for Authors
The text contains many inaccuracies. Some concepts are expressed incorrectly
Reply: Thank you for your words. We agree that the manuscript requires improvement and we thank you for the opportunity to manage them.
In paragraph 55-56 it is wrong to say that there is no limit to the depth of cut and the type of cut : you can choose the depth and the pattern of cut.
Reply: Thank you for your comment. We agree that it may be confusing we changed it to “depth and pattern of the cut can be selected”
Table 1 in paragraph 82 shows the results, not the laser setting values.
Reply: Thank you for this excellent point We have added a table 1 with parameters
In paragraph 84, the laser setting parameters and all values ( diameter, cutting depth, cutting type, diameter) are completely missing.
Reply: Thank you for this excellent point We have added a table 1 with parameters
Thickness of donor’s corneal flap |
220-370 µm |
Thickness of recipient’s corneal flap |
200-350 µm |
Diameter of corneal flap |
7,3-8,2 mm |
Side-cut angle |
850-1000 |
Diameter of stromal bed |
7,0-7,8 mm |
Overlap |
0,2-0,6 mm |
The duration of postoperative therapy is not specified.
Reply: Thank you for this excellent point. We added this part of a sentence.
Topical antibiotic drops (Vigamox, Alcon, Forth Worth, Tx, USA)were administered, Tears Naturale (Alcon), and topical 0.1% Dexamethasone (Polfarma, Warszawa, Poland) slowly tapered over 3-4 months. Sutures were removed during the follow-up controls in patients with high astigmatism exceeding 4.0 diopters starting from 9 months post-op, and hybrid contact lenses were tried.
In paragraph 186-200 it's wrong "the complication connected with putting air into the anterior chamber"
Reply: Thank you for this excellent point. We have removed this part of a sentence.
Reviewer 2 Report
The manuscript summarized the clinical outcomes of femto laser ALK with more than 60 months follow-up. The manuscript described the clinical measurements. It is important issue but some revisions are needed.
1. Some guide comments are included in the manuscript. (Subheading~) Revision is needed.
2. Please review the previous researches of femto laser ALK, especially, comparison studies. Summarize in a table.
3. Can the authors find the risk factors for fast graft failure?
4. Case examples with AS-OCT or slit-lamp images are needed.
5. Can the authors describe the detailed VISUMAX laser settings?
Author Response
RESPONSES TO REVIEWER 2’S COMMENTS
Some guide comments are included in the manuscript. (Subheading~) Revision is needed.
Response: Thank you for allowing us the opportunity to strengthen this manuscript. We have addressed all comments. The changes in the manuscript are highlighted in red.
- Please review the previous researches of femto laser ALK, especially, comparison studies. Summarize in a table.
Reply: Thank you for this excellent point We have added a table 4 with comparison of previous research
Comparison of FALK against |
Results in FALK group |
Number of eyes |
Big-bubble technique in keratoconus[26] |
Worst BCVA and contrast sensitivity |
26 |
Penetrating keratoplasty in keratoconus[27] |
Better BCVA in patients |
24 |
Predescemetic lamellar keratoplasty for keratoconus[28]
|
BCVA, CCT, and ECD insignificant, worst mean refraction |
20 |
Manual DALK[29] |
lower rejection rates, no difference in endothelial cell count between groups |
251 |
Manual DALK[30] |
better UCVA, BSCVA, and refractive astigmatism |
194 |
- Can the authors find the risk factors for fast graft failure?
Reply: Thank you for this excellent point We found no significant risk factors for graft failure when we tried Cox proportional Hazard. We belive that the lack of significance is due to the small sample size of our group.
- Case examples with AS-OCT or slit-lamp images are needed.
We have included fig 2. with the pre and post-op images of the clinical and AS-OCT photos.
- Can the authors describe the detailed VISUMAX laser settings?
Reply: Thank you for this excellent point We have added a table 1 with parameters
Thickness of donor’s corneal flap |
220-370 µm |
Thickness of recipient’s corneal flap |
200-350 µm |
Diameter of corneal flap |
7,3-8,2 mm |
Side-cut angle |
850-1000 |
Diameter of stromal bed |
7,0-7,8 mm |
Overlap |
0,2-0,6 mm |
Reviewer 3 Report
Dear Authors ,
I read with attention the Article on femtosecond laser assisted anterior keratoplasty and here you find some comments and observations
Line 37-41 Please provide bibliography
Line 107-108 You included in the study pellucid marginal degeneration as an indication to ALK procedure. Anyway this ectatic pathology requires eventually, as first line treatment, collagen cross linking or , in more advanced stages, crescentic wedge resection. You didn’t mention these techniques.
Line 121 : remove “of “
Line 132-137: You find a reduction in endothelial cell density How do you explain this loss? Please provide a more accurate discussion on this item
Table 2 : you reported a very High intraoperative complications rate, 50% of incomplete trephination is too high
Line 155: corneal inflammation…what do you mean?
Line 170: frequently? DALK always has advantages over PK in such pathologies, because you leave the endothelium of the patient and have less risk of rejection and less intra and postoperative complications.
Line 173-174: the authors assume that femtosecond laser is able to influence and reduce the postoperative refractive error: how do you think it’s possible to obtain such a result? Don’t you think that the refractive error is mostly due to sutures? What type of suture did you perform in such patients?
Line175: the sentence is incomplete
Line 191: inference without bibliography
Line 194: high corneal swelling due to Femtosecond laser use?
Line 198: the sentence is not closed
Line 200: 50% rate of incomplete trephination is very high, this means that the technique is not reproducible
Line 200-213: are those patients steroid responders? What type of postoperative therapy did you prescribe?
The Authors cited few times the article that Mosca et al. wrote in 2006: at that time femtosecond LK was a new technique and since then a lot of paper on corneal surgery have been published and new advances in dalk big bubble technique have been introduced.
Is it possible that the results you obtained are misrepresented as most of the patients were submitted to manual dissections (that brings stromal scars and postoperative haze)?
Author Response
Line 37-41 Please provide bibliography
Reply: Thank you for this excellent point. We have added a review article by John Dart et al.
Line 107-108 You included in the study pellucid marginal degeneration as an indication to ALK procedure. Anyway this ectatic pathology requires eventually, as first line treatment, collagen cross linking or , in more advanced stages, crescentic wedge resection. You didn’t mention these techniques.
Reply: Thank you for the opportunity to clarify this point.
We have addressed this point in the discussion portion of the manuscript
It is essential to understand that patients with ectatic diseases should have a cross linking or crescentic wedge resection procedure as the first line treatment
Line 121 : remove “of “
Reply: Thank you for this excellent point we have removed the unnecessary off.
Line 132-137: You find a reduction in endothelial cell density How do you explain this loss? Please provide a more accurate discussion on this item
Reply: Thank you for this excellent point we added possible explanation in the discussion:
Another explanation is that endothelial cell loss occurs due to the femtosecond laser disrupting the deep stromal area. Endothelial cell loss depends on the different femtosecond platform used and the settings of the laser[10].
Table 2 : you reported a very High intraoperative complications rate, 50% of incomplete trephination is too high
Reply: Thank you for this excellent point the incomplete trephination happened in 31% it is high but it is what we measured. We had one of the first Visumax in the world (2009) and to say it gently it was not the most fined tuned device ever to hit the market. At the begging we had a lot of technical bugs and issues. My personal opinion is that this laser is great for SMILE only. It is not suitable for transplants as the incomplete trephination is common especially if you compare it to Zimer Z9.
Line 155: corneal inflammation…what do you mean?
Reply: Thank you for the opportunity to clarify this point.
It was a herpetic inflammation.
Line 170: frequently? DALK always has advantages over PK in such pathologies, because you leave the endothelium of the patient and have less risk of rejection and less intra and postoperative complications.
Reply: Thank you for this excellent point we agree that the word “frequently” is wrongly used as the advantage over PK is evident.
Line 173-174: the authors assume that femtosecond laser is able to influence and reduce the postoperative refractive error: how do you think it’s possible to obtain such a result? Don’t you think that the refractive error is mostly due to sutures? What type of suture did you perform in such patients?
Reply: Thank you for this excellent point we agree with you, we use continuous sutures. We have included a figure 1. Regarding the astigmatism, the use of FALK is generally believed to provide lower post-op astigmatism compared to PKP as is presented in papers by Chamberline et. al. or Wade et. al. Naturally we believe that FLAK can reduce astigmatism compared to PKP still it is influenced by the sutures.
Figure 1.1. The procedure commenced with the placement of the donor tissue into an artificial anterior chamber, 2. that was then moved under the laser’s head. 3. After the laser had created a corneal lenticule, the flaps were separated from the rest of the corneoscleral donor tissue using a blunt spatula., or if necessary, corneal scissors. 4. The recipient corneal bed was then prepared in the same way.
Line175: the sentence is incomplete
Reply: Thank you for this excellent point we have added the following sentence and a reference:
However, problems with astigmatism and ametropia are still present in lamellar and penetrating procedures due to the use of sutures
Line 191: inference without bibliography
Reply: Thank you for this excellent point we have added the following publication as a reference:
Femtosecond Laser Assisted Lamellar Keratoplasties
- 5772/20177
Line 194: high corneal swelling due to Femtosecond laser use?
Reply: Thank you for the opportunity to clarify this point.
We believe that the reason of swelling was due to the procedure itself as it is common
We added the following line in the discussion:
Corneal swelling is a common complication after femtosecond procedures[19]
Line 198: the sentence is not closed
Reply: Thank you for this excellent point we have changed this sentence:
The most typical complications, are micro-and macro-perforations and occur infrequently
Line 200: 50% rate of incomplete trephination is very high, this means that the technique is not reproducible
Reply: Thank you for the opportunity to clarify this point. We are sorry for the confusion but as it is stated in table 3 the incomplete trephination was 31 %
Line 200-213: are those patients steroid responders? What type of postoperative therapy did you prescribe?
Reply: Thank you for the opportunity to clarify this point. As we described in the methods sections used: Topical antibiotic drops (Vigamox, Alcon, Forth Worth, Tx, USA)were administered, Tears Naturale (Alcon), and topical 0.1% Dexamethasone (Polfarma, Warszawa, Poland) slowly tapered over 3-4 months Lines 102-104
The Authors cited few times the article that Mosca et al. wrote in 2006: at that time femtosecond LK was a new technique and since then a lot of paper on corneal surgery have been published and new advances in dalk big bubble technique have been introduced.
Reply: Thank you for this excellent point. We have added several new citations including by Chamberlain and Wade and removed the mentions by this author.
Is it possible that the results you obtained are misrepresented as most of the patients were submitted to manual dissections (that brings stromal scars and postoperative haze)?
Reply: Thank you for this valid comment. This might have indeed be the reason, we have included this as the limitation.
Round 2
Reviewer 1 Report
Line 101: it’s wrong syllable synonym of the word recipient
Figure2 change OCT image
line 133 and 153: it isn’t table 1 but table 2
line 175: intraoperative and postoperative complicantions: there arent’t postoperative complications.. if there are you have to write, if there arent’t you have to remove postoperative complications from the title
line 194-196: I don’t understand the context oh that comment
line 215-217: femtosecond anterior keratoplasty ……. Than penetranting keratoplasty : it’is not true
Author Response
Thank you for this time and your effort below please find our point to point
Line 101: it’s wrong syllable synonym of the word recipient
Thank you for this comment we changed it to reci-pient.
Figure2 change OCT image
Thank you for this excellent point we have changed the OCT to make it more visible.
line 133 and 153: it isn’t table 1 but table 2
Thank you for this valuable input we have changed table 1 into table 2.
line 175: intraoperative and postoperative complicantions: there arent’t postoperative complications.. if there are you have to write, if there arent’t you have to remove postoperative complications from the title
Thank you for this excellent point we agree that we did not report any postoperative complications so we have removed it.
line 194-196: I don’t understand the context oh that comment
Thank you for this point, this sentence was part of the template from the journal website we have removed it.
line 215-217: femtosecond anterior keratoplasty ……. Than penetranting keratoplasty : it’is not true
Thank you for this excellent point we have removed this sentence.
Reviewer 3 Report
I found the new version of the Article by Wylegala et al. more suitable to pubblication. The Authors added some technical details related to the laser settings and surgical methods, they also reported on other available surgical and parasurgical techniques to approach the pathologies of the anterior corneal layers and clarify the bias and limitation of the study.
Author Response
I found the new version of the Article by Wylegala et al. more suitable to pubblication. The Authors added some technical details related to the laser settings and surgical methods, they also reported on other available surgical and parasurgical techniques to approach the pathologies of the anterior corneal layers and clarify the bias and limitation of the study.
Reply: Thank you for your kind words and for noting the value of our findings. Further, your help in making our manuscript better is greatly appreciated.